# The Alanine World Model for the Development of the Amino Acid Repertoire in Protein Biosynthesis

**DOI:** 10.3390/ijms20215507

**Published:** 2019-11-05

**Authors:** Vladimir Kubyshkin, Nediljko Budisa

**Affiliations:** 1Department of Chemistry, University of Manitoba, Dysart Rd. 144, Winnipeg, MB R3T 2N2, Canada; 2Department of Chemistry, Technical University of Berlin, Müller-Breslau-Str. 10, 10623 Berlin, Germany

**Keywords:** amino acids, genetic code, RNA World, Protein World, alanine, proline

## Abstract

A central question in the evolution of the modern translation machinery is the origin and chemical ethology of the amino acids prescribed by the genetic code. The RNA World hypothesis postulates that templated protein synthesis has emerged in the transition from RNA to the Protein World. The sequence of these events and principles behind the acquisition of amino acids to this process remain elusive. Here we describe a model for this process by following the scheme previously proposed by Hartman and Smith, which suggests gradual expansion of the coding space as GC–GCA–GCAU genetic code. We point out a correlation of this scheme with the hierarchy of the protein folding. The model follows the sequence of steps in the process of the amino acid recruitment and fits well with the co-evolution and coenzyme handle theories. While the starting set (GC-phase) was responsible for the nucleotide biosynthesis processes, in the second phase alanine-based amino acids (GCA-phase) were recruited from the core metabolism, thereby providing a standard secondary structure, the α-helix. In the final phase (GCAU-phase), the amino acids were appended to the already existing architecture, enabling tertiary fold and membrane interactions. The whole scheme indicates strongly that the choice for the alanine core was done at the GCA-phase, while glycine and proline remained rudiments from the GC-phase. We suggest that the Protein World should rather be considered the Alanine World, as it predominantly relies on the alanine as the core chemical scaffold.

## 1. The Amino Acid Code

Life on Earth is made with the help of biopolymeric molecules of different kinds: linear polymers such as nucleic acids, proteins, and branched chains of polysaccharides. The process of life is characterized by a diverse set of interactions and strong dependencies between these molecules. For example, the synthesis of a cellulose (polymeric carbohydrate) is made by cellulose synthase (protein), a stretch of DNA (nucleic acid) is made by DNA polymerase (protein), and any full-length protein is made on the ribosome, which is essentially composed of RNA (nucleic acid). The relationships become even more complex considering that some of these syntheses are templated: a DNA is made on another DNA template (this process is called replication), an RNA is made on a DNA template (transcription), and a protein is made on an RNA template (translation). A templated synthesis implies that the sequence of the source polymer corresponds to the sequence of the outcome polymer with a certain rule. As a result, the source polymer can be considered as the one containing information about the outcome polymer’s content and its properties and the latter essentially execute the information. Therefore, a DNA (or RNA) is also called an informational polymer, whereas a protein is called an executive polymer. 

What are the constraints in the choice of basic building blocks (monomers) for these transient polymeric structures? In other words, to what extent was monomer appearance inevitable for life on Earth and how likely is it to expect this component to appear in a living organism elsewhere in the universe [1]? This question is especially frequently asked with regards to amino acid composition [2]. There are 20 + 3 amino acid residues that are templated in the ribosomal synthesis of proteins (coded into proteins). How did these amino acids end up in the protein code is a controversial matter that has been addressed via the analysis of the codon structures, RNAs, metabolic and prebiotic chemistries, physiochemical properties of amino acids, and other chemical and physical perspectives [3,4,5,6]. While questioning the amino acid repertoire, we tend to overlook that the amino acid identities represent only one aspect of the protein architecture: the primary sequence. There is however more to that: secondary structures (e.g., α-helix and β-sheet), motifs (hydrophobic, salt bridges, etc.), and tertiary folding (globular fold and membrane proteins) (Figure 1). Their appearance in protein architecture is rarely discussed or questioned. Meanwhile, this is exactly what the amino acid sequence is supposed to determine. Why do we not turn our perspective and ask ourselves: is it inevitable that proteins should span the membrane when making channels, pumps, or receptors? Or another question: does the α-helix/β-sheet pair have any alternatives in forming the main protein constituting element? Any kind of answer will provide an absolutely new and unique view on the set of the amino acids involved in protein biogenesis. In this article, we will use this perspective to propose a view on the amino acid repertoire establishment. 

## 2. Amino Acids Do Not Equal Proteins

Amino acids play multiple metabolic, energetic, and structural roles in biochemistry, and for many of them participation in protein translation accounts for a rather small part of their cellular activity. This implies that amino acids should have entered biochemistry at least twice: first, as metabolic entities, and later, as monomeric constituents for ribosomally produced proteins. In between was the RNA World. The hypothesis of the RNA World follows directly from the Central Dogma of molecular biology, and it postulates that there was a time, when biochemistry was driven with the catalysis of RNA molecules [7,8,9]. Whether there were any protein molecules in the RNA World phase of before, we do not know. We cannot know this for sure, and will remain agnostic regarding the molecular components that are not trackable in the existing biochemistry. Only those proteins that were produced by the ribosome are coded by genes, thus they can be phylogenetically analyzed at the present days. This suggests that protein biogenesis and the canonical amino acid repertoire have emerged and evolved in the RNA world phase in the form that was retained until present day. This hypothesis provides much information about the amino acid repertoire set in the Protein World. In our narrative, we will start from RNA hypothesis and try to derive the amino acid repertoire with the help of some additional premises. This retrospective analysis will eventually lead us to the Alanine World model.

We do acknowledge that there are suggestions regarding the presence of peptides in the pre-coded phases [10]. However, we are very skeptical towards speculations about the development phases prior to the Protein World, especially when these are based on the analysis of the genetically encoded proteins. A typical example is the Thioester World hypothesis [11], which suggests formation of peptides before their ribosomal synthesis. Perhaps, the thioester world should be functionally similar to present day non-ribosomal peptide syntheses, which is performed by relatively large and cumbersome protein complexes [12]. To criticize this view, we would like to note that the Thioester World hypothesis is primarily based on side reactivity modes of aminoacyl-tRNA synthetases (aaRSs) and analysis of existing protein sequences [11]. However, it was never explained how these proteins could exist before they were encoded by an mRNA. Indeed, many believe that aaRSs and other ribosomal proteins are among the oldest proteins, and their analysis can deliver valuable information about evolutionary past, for example, on how some principles of the protein fold and reactivity were developing [13,14]. However, if a hypothesis suggests existence of aaRSs before the Protein World (before they were coded by the mRNA), this means that the information somehow reverted from protein to nucleic acid. At the same time, there is no mechanism by which any hypothetical pre-ribosomal protein could encode itself into a nucleic acid, as such an event is directly forbidden by the Central Dogma [15]. Any suggestion that proteins existed before coding, and then these ‘somehow’ ended up in the mRNA generates a severe logical contradiction: why do we accept the RNA World hypothesis from the Central Dogma, but neglect that the Central Dogma literally forbids any protein-to-RNA and protein-to-protein information flow? The question is of course rhetoric. For that reason, we believe that current phylogenetic analyses have rather limited significance in understanding of the phases prior the mature Protein World, before the genes received their complete protein meanings via the amino acid assignment. Thus, speculations based on chemical logic can be much more informative than formal and rather speculative positivistic analyses of existent genes or protein sequences. 

Another common misconception, which we would like to criticize, is drawing a direct connection between amino acids and proteins. Amino acids do not equal proteins. They play a multitude of roles, and derivation of polypeptides from amino acids is justified as much as deriving nucleotides (Figure 2) or metabolic products (e.g., oxaloacetate from aspartate). Peptides do not equal proteins either. Any formation of a stochastic di-, tri-, or oligopeptide in prebiotic or rare biochemical transformations, does not automatically assume existence of proteins, polypeptides with defined sequences, 3D structures, and functions. 

Our argument does not imply that the amino acids were not present in the RNA World and before, just the opposite. As a matter of fact, they were available for the outbreak of the protein biogenesis, and for that reason they should have been there in the RNA bound form. There is no way to avoid the use of such simple and abundant structures as amino acids, once biochemistry started. This is why we assume that the amino acids entered the biochemistry long before they started making proteins, and their initial use was different. The primary and essential role of the amino acids was involvement in metabolism. For example, in order to produce RNA molecules, there is a definitive need for some amino acid moieties. Nucleobases are inherently related to amino acids, as both are composed out of C/N/O/H atom combinations. In fact, all nucleobases are produced with the massive involvement of amino acids as the precursors (Figure 2). Some of those come to nucleobase heterocycles in complete pieces: glycine for purines and aspartate for pyrimidines. Thus, if we would consider that the RNA World existed for at least some time, we should expect the presence of amino acids involved in production of the RNA building blocks, nucleotides. While it has been suggested that the first synthesis of the nucleotides was performed on mineral surfaces [16], its potential transfer to the RNA bodies should require loading of the amino acid to the RNA bodies, which effectively lead to the aminoacyl-RNA esters. 

The multi-phase entrance of amino acids in biochemical processes perfectly explains why the set of amino acid available in the mineral world (pre-biotic chemistry or meteorites) [17,18,19] does not align the set of 20+ proteinogenic amino acids. In fact, there is no need to align them. Neither, there is a need to require each mineral amino acid to be available in the form of one enantiomer. The amino acids had to be supplied metabolically because otherwise the amino acid could not be retained in continuous translation process. In the first phase, mineral amino acids (chiral or not) were probably recruited to launch simple and basic biochemical processes, since these are among the easiest and most primitive organic molecules needed to construct more complex matter. In the second phase, (chiral) amino acids were recruited (from metabolism) to start making polypeptide polymers. These events need to be separated, as logically follows from the RNA world hypothesis. The amino acids in the proteins are not necessarily the amino acids that fueled emergence of life in the first place. Though, the proteins are still at the core of biochemistry as we know it and the set of 20+ amino acid constituting proteins settles the boundaries of our Earth’s biochemistry. This is why we should understand how this set was formed.

## 3. The Evolution of the Genetic Code

The history of the protein biogenesis is the history of coding. There are 20 amino acids that are universally coded to the proteins, each having a correspondence to an element in the coding RNA (DNA) sequence. The coding units are called codons, and the correspondence between them and the amino acids in the polypeptide sequence is called the genetic code. In biochemistry, the genetic code does not mean what is written, but how to interpret what is written—the rule of decoding. However, this rule is not simple, as it involves 20(+3) amino acids and 64 (61 + 2 coding) different codons, combinations thereof, and a complex way of decoding [20]. One of the common theories [21,22] suggested that the coding (i.e., association between codons and amino acids) was arbitrary once started, and then probably developed via an expansion. Then it reached a point, when no further expansion was possible, thus the code was settled or “frozen”, and this is the genetic code as it is now [23,24]. How did this development go?

Hartman and Smith recently summarized one possible scenario of this development in a very simple and elegant scheme [25,26]. It suggests that the first amino acid set for life on Earth was a GC code, which means that the coding stretch was only composed out of these two nucleobases: guanine and cytosine. As such, the messenger RNA code could make a sequence made out of four amino acids: glycine, alanine, proline, and one amino acid with a positively charged side-chain, which we suggest to be ornithine (now: arginine). The presence of the positive charge was essential for interaction with polyanionic RNA, thus it enabled molecular mechanisms for subsequent co-evolution of peptide sequences and translation apparatus. Once started, the repertoire of the amino acids underwent an expansion. It is then hypothesized that the next evolutionary step in the code was the addition of the A letter to the messenger RNA, which implies the acquisition of polar amino acids resulting in a GCA code. The amino acids, aspartic and glutamic acid, asparagine and glutamine, threonine, serine and histidine, were added on this stage of development, each being just a few steps away from the central metabolism. In the final step, the GCAU-phase, the addition of the U letter allowed the acquisition of the hydrophobic amino acids, methionine, leucine, isoleucine, valine, phenylalanine, tyrosine, tryptophan, as well as cysteine and lysine, all requiring relatively complex biosynthetic pathways.

## 4. The Repertoire Expansion: Step by Step

The GC–GCA–GCAU evolutionary scheme was originally derived by Hartman and Smith from the analysis of the aaRSs [25,26], but it correlates well with the complexity of the tRNA modifications [27] and metabolic relations between of the amino acids in their biosynthetic pathways. The latter has also been suggested in the co-evolution theory, which considers the expansion of the amino acid repertoire alongside the development of the metabolic capacities of the primitive organism [28]. Even more exciting is to note that the GC–GCA–GCAU scheme implies co-evolution of the hierarchy of protein folding [29,30]. 

To explain this argument, we would like to analyze the meaning of this metabolic scheme with regards to the protein structure. The set of amino acids in the original phase (glycine, alanine, proline, and a cationic one) is not characteristic to most of the usual proteins, except perhaps collagen [31]. However, once we agree that collagen was a much later evolutionary invention [32], we should face the fact that the primitive polypeptides made out of this set were barely functional. The reason for this is clear: such peptides are not competent of forming strong and long-lasting secondary structures. In the modern biochemistry, proline and glycine are known as breakers and kink inducers, when these appear in the context of most common α-helical or β-strand regions. If half of the repertoire is ‘breakers’, a proteome made out of this set will not be robust. Besides, the only chemical function available in this proteome is one positive charge from ornithine (now: arginine), which is enough to complement the peptides with the negatively charged nucleic acid bodies [33], but this is certainly not enough to build-up any significant protein complexity. The repertoire underwent a subsequent expansion. Evolution made it probably in the simplest and most straightforward way possible: it acquired what was available. The amino acids recruited at the GCA-phase are direct derivatives of keto acids from the tricarboxylic acid cycle, the core cycle of biochemistry [34]. The only exception is histidine—a derivative of a nucleotide processing branch, its production in the RNA World should have been quite likely considering the availability of the nucleotide precursors.

Here comes an interesting observation. The common trait of the amino acids acquired in the GCA-phase is that they are all related to alanine. All these structures constitute four fundamental elements: a defined chirality (absent in glycine), a hydrogen bond donor (absent in proline), a hydrogen bond acceptor, and a substituent placed at the β-carbon atom (and not attached through a distant amino-group like in arginine). The alanine core forms the backbone of the peptides, thereby enabling α-helix and β-strand structures to be formed (Figure 3). From this point on, the α-helical architecture dominated the protein biochemistry. The significance of the GCA-phase is that the recruitment of alanine derivatives at this phase allowed the primitive organism to come up with the α-helical secondary structure standard, and this boosted further development. For example, if another organism would recruit different types of residues at this phase, a proteome development would be more complicated due to the lack of a structure standard, and the evolution would be severely hampered. Therefore, the acquisition of the alanine derivatives in the GCA-phase yielded a clear advantage. To avoid confusion, we would like to stress that by ‘alanine derivatives’ we refer to amino acids that share the same core chemical structure, and not metabolic derivatives. 

After the standard was settled, the complexity development went on, but it still lacked the most exclusive component of protein structures, the hydrophobic motif. The hydrophobic motif is a true invention of the protein chemistry, and it has no equivalents in any other biopolymer. Not only does the hydrophobic motif cause tertiary folding according to the ‘*polar out–apolar in*’ principle, it allows proteins to go into the membrane (‘*apolar out–polar in*’ principle). Only proteins can operate at the membrane, it is their completely exclusive domain among biopolymers. The conquest of the membrane is one of the most outstanding events, and its significance cannot be overestimated: if there is no membrane there is no cell! 

The entrance of the membrane would not be possible without the α-helical standard. Let us consider that sugars are very polar structures, nucleic acids are extremely polar too, but so are the peptides. This is due to the high polarity of the amide groups constituting the peptide backbone. Any addition of hydrophobic side-chains to the polar amide skeleton generates essentially ambivalent structures, as there will be an opposition between the backbone and the side-chains. This is why, it is a very challenging task to reduce the ambivalence and construct peptides able to go to the core of a lipid membrane, the most unfriendly environment for a polar structure of a peptide. This cannot be done without a stable underlying backbone conformation, a reliable secondary structure skeleton. Fortunately, the dominant secondary structure, the α-helix, was already invented in the GCA-phase, so the addition of the hydrophobic residues was done to the α-helical chemical basement. This was done in the GCAU-phase, where addition of the U letter enabled influx of the amino acids known as hydrophobic. The latter definition needs to be critically evaluated. The side-chains can indeed be hydrophobic, whereas the amino acid part, which is essentially derived from alanine, still remains exceptionally polar. The ‘hydrophobic’ amino acids should rather be called ‘amphiphilic’, but they form a hydrophobic motif when they appear in the α-helical context of a peptide structure. This is due to the fact that the side-chains cover up the rigid polar backbone core, and prevent its solvation by the environment, as the result the whole peptide appears as hydrophobic. The lack of the backbone exposure, and not the hydrophobicity of the side chains, is what creates the hydrophobic motif. Leucine does this job best in soluble proteins [35], whereas β-branched valine and isoleucine act as better insulators when an α-helix needs to be immersed into a hydrocarbon core of a lipid membrane [36]. The hydrophobic motif relies on the stability of the underlying α-helix: an unstable secondary fold will lead to an exposure of the polar backbone to the environment, thereby the hydrophobicity feature will be compromised. This is why the compact α-helical architecture has to be kept stable. This requirement forms requirements to the structures of the ‘hydrophobic’ (amphiphilic) amino acids. For example, it limits branching of the amino acid structures at the β-position (see Figure 3): one hydrogen substitution is allowed (phenylalanine, leucine), two hydrogen substitutions—disfavor the helix except when these are in the membrane milieu (valine, isoleucine), and three—destroy the helix (*tert*-leucine). The set of the ‘hydrophobic’ coded amino acids follows this limitation. 

It should be noted that the protein interaction with the membrane is the first and only possible non-covalent interaction of biopolymers with the lipid environment. Neither sugars nor nucleic acids can be sufficiently hydrophobic to penetrate, cross or function in hydrophobic milieu of the membrane core [37]. Thus, the presented model goes in line with the theories that predict late encapsulation of the biopolymeric molecules to lipid vesicles [7] following the ‘RNA first’ argument [38]. The proteins should have bridged the gap between the RNA and the Lipid Worlds due to the hydrophobic elements and their ability to carry a positive charge, thereby enabling indirect interaction between the anionic RNA and anionic/zwitter-ionic phospholipids. In this way, the latest GCAU-code phase enabled subsequent co-evolution of the lipid system with the biopolymeric components [37]. 

## 5. The Alanine World

The GC–GCA–GCAU scheme proposed by Hartman and Smith is very appealing for one very important reason: it makes sense. The correlation with the hierarchy of the protein folding is remarkably straightforward. Our model suggests that the metabolic processes, principal protein folding elements, and the protein synthesis apparatus co-evolved. Because this happened as a result of a random walk, the polypeptide synthesis was not an intended biochemical process when it started. First, amino acid polymerization occurred as an accident (or side-process), therefore it acquired a rather casual set of amino acids (GC-phase) that was suited for the RNA structures rather than proteins (see Section 7 below). Once the polypeptides showed their beneficial action (Figure 4), the translation apparatus recruited more amino acids from what was available (GCA-phase). At this point, the expanded repertoire featured formation of the α-helix. In the last phase (GCAU-phase), more complex structures were recruited, and this enabled the tertiary fold and interactions with the membrane (Figure 4). The hydrophobic residues of relatively complex metabolic origin were added to a ready secondary structure set, therefore they were derived from the alanine core. As a result, the expansion of the repertoire went solely into the direction of alanine derivatives. The resulting genetic code contains essentially three types of residues: glycine, proline, alanine, and 17 other structures with derivations at the β-atom of alanine. Thus, we propose the term Alanine World as more specific than Protein World in describing the life as we know it on Earth.

Analysis of the GC–GCA–GCAU scheme shows that the most common catalytic residues were appended to the genetic code in the GCA-code. Among them, serine/threonine, aspartate/glutamate, and histidine are the most prominent. Interestingly, it has been suggested that the transit from the RNA to the Protein World had an intermediate phase with peptide/RNA hybrid molecules doing catalysis [39] (also proposed as the Peptide/RNA World [40]). The Alanine World model agrees well with this hypothesis, and suggests that the hybrid phase actually was the GCA-phase. In this phase, short peptide stretches performed catalytic reactions while still bound to RNA, and the RNA provided the overall folding and the geometric arrangement for the catalytic residues. Subsequent emergence of the hydrophobic motif in the GCAU-phase enabled the globular fold (‘*apolar in*’). As the result, the RNA and the Protein Worlds were ultimately separated. 

The appearance of the structure standard in the Alanine World has numerous consequences in modern biochemistry. Firstly, most of the amino acids can be interchanged by point mutations, while the secondary structure still remains intact. The fact that alanine mimics the secondary structure preferences of the majority of coded amino acids is exploited in Alanine-Scanning Mutagenesis, a laboratory method that helps to determine the ‘importance’ of a particular amino acid side-chain in the functions of a protein [41]. In this method, a residue is mutated to alanine (and not to proline or glycine!), thereby the side-chain functional group is removed while the backbone folding typically remains constant. The interchangeability of the alanine-derived genetic repertoire can also be seen in the fact that homologous proteins share only some part of the sequence identity (e.g., 90%, 80%, 70%, or even less), while still maintaining the same fold and functions [42]. This could not be possible without the common structure standard. Another consequence from the shared alanine architecture is the letter approximation. In the RNA and DNA sequences, we assume these to be composed out of ‘letters’ due to the fact that there is same backbone, and only the nucleobase identify differs between the individual nucleotides. The same approximation is transferred to protein sequences that are also considered to be composed out of 20 amino acid ‘letters’. This is due to the fact that the amino acids share the same common architecture, the alanine core, with only the side-chain substituents varying. In the case of a more diverse backbone, for example, involving secondary, β-, or d-amino acids, for each sequence combination there would be a complex pattern of the backbone structures and the side-chain placements in space. As the result, it would be very difficult to create the alphabetical abstraction, and one would have to consider the actual chemical constitution instead. 

Does the Alanine World have alternatives? From the standpoint of peptide chemistry, it does. Potentially, the derivation of the side-chain structures could be done at different places (Figure 5). For example, if we assume that the original amino acid in the GC code was ornithine, then the derivation of this structure with a guanidinium group forms arginine. Following the same principle, the terminal amino-group in lysine can be decorated with a chemical function (Figure 5). Such design principle is implemented in pyrrolysine, a special coded amino acid, which occurs in some methanogens [43]. We may suggest that the derivation of the distant amino-group should open an opportunity to form a diverse proteome, but in the existing biochemistry, this development only has a few rudiments: arginine (universally coded) and pyrrolysine (coded in special organisms only).

Similarly, if we envision formation of the proteome from the glycine structure directly, we could draw many possibilities (Figure 5). The glycine structure is so simple and generic that discussing different derivation options could severely exceed the length of one article. Here, we would like to mention peptoids, peptide structures, where the side-chain is placed at the nitrogen rather than carbon atom in the backbone [44]. Peptoids have been studied in applied chemical research, but in biochemical settings these structures could exhibit many interesting features. For example, they lack the H-bond donor site, and they do not necessarily have a chiral center in their structure, as such the chirality features of the backbone folding (e.g., helix handedness) can be a flexible element. Another notable class of peptides is peptaibols, the structures with two substitutions at the α-carbon. The simplest structure in this set 2-aminoisobutiric acid, which favors the α-helix, whereas larger substituents can help stabilize other backbone conformations such as 3_10_-helix [45] or fully extended 2.0_5_-helix, an unusual structure not present in natural proteomes [46]. 

Overall, we should note that any manipulation with the core amino acid element, and exchange of alanine with something else may lead to drastic difference in the secondary structure set, the stability of the secondary structures, their typical length, and lifetimes. Nonetheless, the secondary structure part should still form tertiary fold and even membrane spanning elements, when properly decorated with motifs. The concept of an exchanged secondary structure can best be illustrated with the Proline World. 

## 6. The Proline World

In the Earth’s biochemistry the major secondary structure in proteins is the α-helix followed by the β-strand. The third one is the polyproline-II helix [47]. This name should not leave an impression that the structure is specific to proline only. The polyproline-II helix is ubiquitous, it can be adopted by all coded amino acids, especially when the α-helical structure cannot be formed for certain reasons (e.g., due to the presence of denaturing agents) [48]. As the most generic structure, this was probably very abundant in the earliest GC-phase of the proteome evolution, before α-helix took over this role in the GCA-phase.

Though, the name of the structure indicates relationship with proline due to the fact that it was discovered in polymeric proline [49,50], and this happened for a reason. The proline structure lacks a hydrogen bond donor site (N–H), which precludes formation of α-helical or β-strand structures. This is however, not unique, and also observed with other N-alkylated amino acids (e.g., sarcosine). A more special property of proline is that this residue is cyclic and forms a five-membered ring. The ring structure of proline has the consequence that the number of molecular arrangements in space, molecular conformations, becomes limited (Figure 6). As a result, polymeric proline can adopt only a few types of conformations [51]. Polyproline-II is one of the backbone conformations featured by proline, and this is the dominant structure for this scaffold in water. In nonpolar solvents, another helix becomes dominant, the polyproline-I helix. These helices have very distinct geometric properties, and the transition between them is mainly driven by the polarity of the medium and the temperature [52,53,54]. However, it can also be manipulated with the substitution pattern in proline [55,56]. Moreover, it was recently shown that the oligoproline scaffold can also adopt a special type of a β-structure in case of certain chemical analogues [57,58].

Thereby, the chemical scaffold of polyproline has its own specific conformational variability, and could constitute a few distinct conformations as underlying structures for motifs. Substitutions on proline can be afforded at seven distinct places (compare to just three—in glycine or alanine!), resulting in alteration of the conformational propensities. Nature has explored this option, although it did so relatively late in the evolution. The most common is hydroxylation that occur in collagen yielding hydroxyproline [59]. Other modifications include alkylations at the proline ring [60].

What if these modifications were available already in the GC-phase? We envision that this would open up a way towards building a peptide complexity based on the proline structure, and lead to a whole new world as the result, the Proline World. The Proline World is the world where the proteins are coded to primary sequences, the sequences contain patterns of hydrophobic/charged/etc. residues (through derivations at the proline ring), motifs determine the 3D folding of proteins—everything just like in the Alanine World, but the secondary structures and related distances and periodicities are different, and rely on the conformational transitions of polyproline. It is quite important to note that Proline World is inclusive since polyproline-II helix is one of the most generic folds in nature. Most α-amino acids (alanine derived or not) can adopt the polyproline-II structure too, although, with lesser stability [61]. For this reason, it is possible that other amino acid types could be easily tolerated in the Proline World.

The GC–GCA–GCAU scheme discussed above suggest that the polyproline-II type of folding was widely present in the initial (GC) phase before it lost the competition to the α-helix (in the GCA-phase). The most critical question though, is this structure competent of building a hydrophobic motif and go to the membrane? The answer is ‘yes’. Very recently we demonstrated experimentally that the polyproline-II helix can be integrated to the membrane and adopt a defined transmembrane alignment [62,63]. A successive study showed that the collagen helix, a fold made out of assembled polyproline-II helices, has no fundamental limitations to do so as well [64]. As a result, we suggest that there should be a way to build up a membrane protein complexity, which entirely bypasses any involvement of an α-helix. Thereby, we provided experimental basis for the consideration of the Proline World as a fully realistic scenario that is alternative to the Alanine World. 

Overall, this makes the Proline World a conceivable option for an alternative way of building life. This is probably not obvious, when we just look at the amino acid structure alone, but it becomes easily conceivable when we consider the secondary fold it builds.

## 7. Why Proline? Why Not Proline? 

To this point, we discussed the different stages of the amino acid recruitment, all except the initial set in the GC-phase. Perhaps the most intriguing question in these regards will be: why would such a bizarre structure as proline be involved in the RNA World chemistry at all? Obviously, it cannot be taken as a precursor for nucleotides (as glycine or aspartate), neither can it serve as a donor of an amino group in transamination processes (like alanine, glutamate, or others), it even lacks the side-chain function groups (no positive/negative charge, etc.). Thus, the only possible remaining role for the proline in the RNA World is involvement in catalysis. If so, there is only one place in the structure of an RNA bound proline, that is capable of catalysis – the pyrrolidine ring (Figure 7).

Exceptional catalytic properties of proline in catalytic processes has been known in the organocatalytic research [65]. They are associated with the secondary amino group enclosed to a five-membered ring together with a chiral center for a chirality induction. The reactions catalyzed by proline, and its C-terminal derivatives are called condensation reactions, the same type of reactions are also believed to be responsible for the production of sugars in the early phases of life formation [66,67]. This led some to the suggestion that proline was the key amino acid involved in homochirality transfer from amino acids to sugars [68]. Here, we do not aim to evaluate the significance of proline in prebiotic processes, but we would like to note that same argument can be applied to the RNA World. The sugar production for the nucleotide biogenesis may perfectly explain the essential involvement of the prolyl-RNA esters in the RNA World processes (Figure 8). The proline catalyzed condensation of sugars offers a clear advantage to a cyanide based scheme. First, proline can be easily bound to a specific (catalytic) site of an RNA/ribozyme, while cyanide cannot. Second, proline contains a chiral element that should lead to a needed chirality induction, which cyanide cannot do. 

Furthermore, the recruitment of proline readily proposes an explanation for the occurrence of ornithine in the same biochemical phase (Figure 7). Ornithine is a precursor towards proline in one of the two major metabolic pathways [69]. Therefore, the production of catalytic prolyl-RNA esters could involve the ornithyl-RNA as its precursor. We would like to point out that according to this scheme, the original intended purpose of the aminoacyl-RNA esters was that they were involved either as precursors or as catalysts in chemical transformations, and not for production of peptides. This suggestion was previously expressed in the coenzyme handle hypothesis [70]. The attachment of the amino acids to the RNA mediators seems to be a valid assumption, because the RNA bodies would have difficulties to operate with otherwise too small amino acid structures, for example, when recruiting them directly from the surrounding media. Following this argument, we think that the chemistry in the RNA World should have been done in a different way to what we have now. For example, instead of operating with the molecules from the cellular pool, an RNA would fish all amino acids from the pool (just like this is done in the experimental ribozyme known as flexyzyme [71]), and then define their structures through the chemical processing. Thus, instead of the key–lock model of the Protein World, the RNA World might assume a different, processivity model (Figure 9). In the key-lock model, an amino acid is recruited by the cognate RNA due to the action of a unique pocket, specific for this only amino acid. As mentioned, this would be problematic in the RNA-only phase because the amino acids are too small. In the processivity model, an amino acid is attached unspecifically, and it remains attached to an RNA as long as it undergoes correct chemical processing. If it does not, the RNA stalls, and the wrong amino acid is hydrolyzed after some time, simply through the action of water. Such mechanisms are utilized in the editing schemes in the aaRSs [72]. In this way, the processivity of ornithine towards proline suggests its presence in the RNA World in the RNA-bound form. Ornithine can be processes towards proline in two steps, where the first step is oxidation of an amino group towards a carbonyl. The same type of reaction is utilized in transamination (see Figure 7), and this should have been present in the RNA World since many of the GCA-phase amino acids are produced this way. Both metabolic neighbors, proline and ornithine, were recruited to the starting proteome. 

The polypeptide synthesis, however, was a rather undesired side process in the GC-phase. Ornithine is evidently compatible with the polypeptide synthesis, and it is involved in non-ribosomal peptide synthesis of some peptides (e.g., gramicidin S or ramoplanin). Nonetheless, it has been pointed out that ornithine might generate truncations due to the likelihood of the side-chain-backbone cyclization [74] (see Figure 7). Such truncations could be tolerable in the GC-phase, where the polypeptides were undesired themselves, but later when the polypeptides have become the intended products, side-chain reactivity would severely reduce fidelity of the polypeptide synthesis by generating multiple truncations. This reveals the reason why ornithine residue was transformed to arginine by adding a chemically inert guanidinium group that maintained the positive charge. The addition of this group completely abolished the reactivity of the side-chain with the backbone [74]. 

The catalytic role of proline in sugar formation might greatly justify the initial set in the outbreak of the protein biosynthesis. If indeed proline was involved in the production of sugars, then perhaps this was the only amino acid, which was initially required to be enantiomeric. In principle, the amino acids involved in nucleobase biogenesis do not have to be chiral, in the course of the synthesis their chiral features will be lost. However, whenever they were produced metabolically (e.g., from the citric acid cycle), the matrix of the catalytic RNA body should produce them in the enantiomerically pure form, simply because biochemistry functions in this way. Some studies have suggested that the biases for specific enantiomers in biogenic molecules results from preferred interactions between sugar and nucleotides with a given chirality, and amino acids with the opposite chirality [75,76]. A typical example of a chirality establishment is transamination of pyruvate, which yields enantiomeric L-alanine (Figure 7). Alanine is a generic structure, and this should have been around for various purposes very early in biochemistry, and it is likely to expect it in the GC-phase. For example, this could have a role as a nitrogen atom carrier, before this was taken over by more complex glutamate and glutamine. Alternatively, an RNA bound alanine could be serving a role as a pyruvate equivalent, the same as it is used in the C4-plants. Finally, the recruitment of glycine in the RNA World chemistry was justified by its involvement in the synthesis of purines. The biogenesis of the glycine is among the simplest ones, and involves a tetrahydrofolate precursor (also involved in construction of purines), carbon dioxide, and ammonia [77]. 

We consider it quite likely, though, that proline played a key role in bridging amino acid and sugar chemistries in the formation of the RNA World. That same amino acid was recruited among first in the transformation to the Protein World was nearly inevitable. However, the recruitment of the metabolic structures based on the generic alanine skeleton in the GCA-phase discarded the Proline World option. Why? Proline derivatives were simply not available. For example, what does it take to make hydroxyproline? Molecular oxygen, a component which was absent in the RNA World phase [78]. As noted above, the aliphatic ring or proline provides seven sites for substitutions, but they are relatively inert and require action of aggressive reactants such as molecular oxygen. In organic chemistry this is called the sp^3^-CH-activation problem. For comparison, the synthesis of tyrosine from prephenate involves the oxidative action of NAD, which is an anaerobic agent. Tyrosine (hydroxy-phenylalanine) is present in the genetic code and hydroxyproline is not (Figure 10).

Metabolic unavailability and synthetic complexity of the proline derivatives did not let the proteome develop in the direction of proline derivatives, and closed the Proline World path. Conversely, the availability of the alanine derivatives directed the genetic code development, and lead to the Alanine World. The diversification of the proline residue in biochemistry occurred much later, when the genetic code was already settled. However, if there was molecular oxygen at the start, such a result will not be that certain. 

## 8. Concluding Remarks

Amino acids have numerous functions in biochemistry, and protein biogenesis is arguably the most complex among them. Ribosomal RNA is the molecular machine, which polymerizes proteins according to the messenger RNA template. The emergence of the protein coding, the transit from the RNA to the Protein World, is the process which determined the repertoire of the amino acids involved in it. However, one should clearly realize that the amino acids, as simple and most basic functional organic molecules, should have been involved in other processes much earlier. When the transit started they were readily available in the activated form bound to RNA, and we can suggest their role as some involvement in metabolic processes as precursors and catalytic auxiliaries (the ‘co-enzyme handle’ hypothesis [39,70]). The RNA World hypothesis readily suggest some involvement of the amino acids in constructing nucleobases or catalyzing chemical transformation such as condensations of sugars. 

Furthermore, the analysis of the GC–GCA–GCAU scheme of the genetic code development proposed by Hartman and Smith allows us to reconstruct the whole sequence of events, as well as the etiology of the amino acid appearance in the repertoire. In our narrative, we assumed that the recruitment of the amino acids was from metabolic and not mineral (e.g. from meteorites) sources, because only this scenario ensures a continuous supply of chiral amino acids. Subsequent modeling suggests that the first set (GC-phase) was the amino acids related to the nucleotide biosynthesis, and included diverse core structures, proline, alanine, glycine, and ornithine (now: arginine). This was followed by recruitment of a set of the amino acids closest to the core metabolism (the GCA-phase). This phase allowed for agreement on a standard core structure, the alanine core, and resulted in the dominance of the α-helix. This formed the Alanine World. The settlement of the standard was a key element, which allowed design and recruitment of the metabolically complex amino acids with hydrophobic side-chains, as well as the rest of the amino acid repertoire featuring tertiary fold and membrane interactions (GCAU-phase). We thus demonstrate that the Alanine World model requires that complex protein fold followed the amino acid repertoire expansion [29,30], whereas some other works suggested the reverse order [79,80]. We also show that an alternative development was possible. In the case when molecular oxygen (or another generic CH-activator) was present in the initial phase, this could enable derivation of proline, and a Proline World alternative would be at least more likely, if not fully possible. Potentially, alternative genetic code developments could be reconstructed artificially in laboratory evolution experiments. For example, by mimicking bioavailability of certain chemicals, one could expect them to be included in the build-up of the biological complexity, and such experiments should be conducted in the future. 

Initial derivation of most amino acids from the alanine core, allowed nature to standardize the underlying secondary structure elements. Results from this can be observed in existing biochemistry. For example, we can consider amino acids by their side-chains identities, and approximate them as 20 ‘letters’ in a similar fashion to the nucleic acid ‘letters’. The concept of biochemical ‘information’ relies on this approximation. The interchangeability of residues, resistance of proteins towards most point mutations (in line with the Kimura theory that most mutations are neutral [81]), and many more features of the proteome result from the fact that the underlying amino acid core remains the same, except for a few rudimentary cases: glycine, proline, and pyrrolysine. We thus propose the Alanine World as a retrospective model that proposes a chemical etiology of coded amino acid repertoire in proteins as a result of a historical process. 

## Figures and Tables

**Figure 1 ijms-20-05507-f001:**
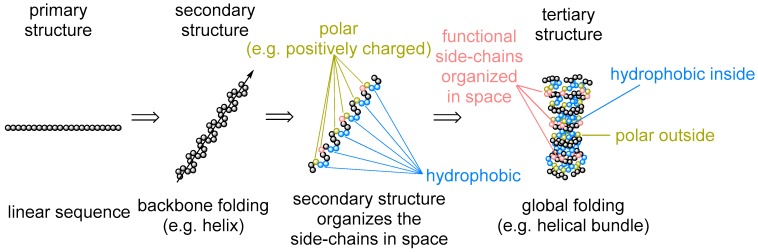
The hierarchy of protein folding.

**Figure 2 ijms-20-05507-f002:**
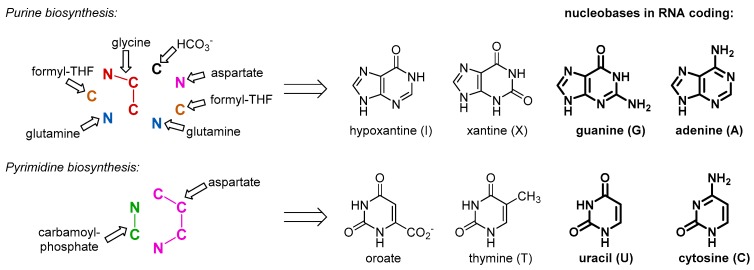
Relationship between the amino acids and nucleobases: all nucleobases are formed in metabolic reactions involving amino acid as sources of the building blocks.

**Figure 3 ijms-20-05507-f003:**
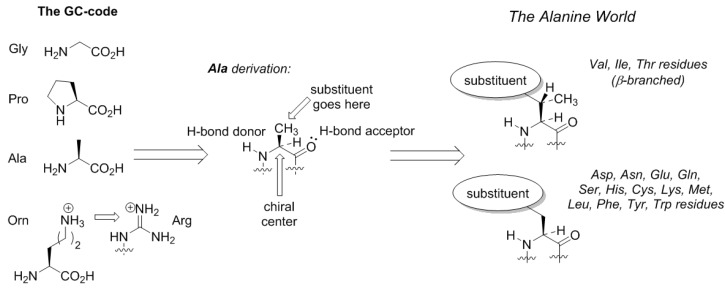
The development of the amino acid structures towards the Alanine World.

**Figure 4 ijms-20-05507-f004:**
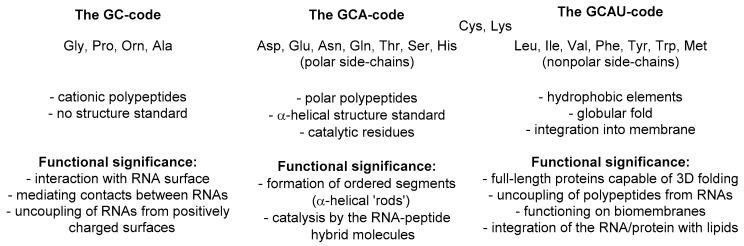
The increasing complexity of the polypeptide structures and functions along the GC–GCA–GCAU development scheme.

**Figure 5 ijms-20-05507-f005:**
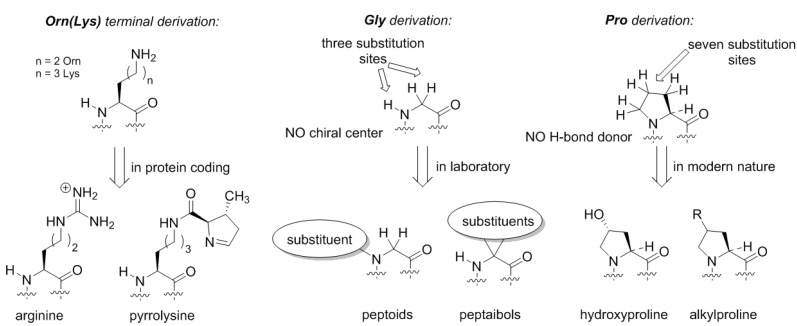
Some alternative backbone developments that have remaining rudiments in the existing genetic code.

**Figure 6 ijms-20-05507-f006:**
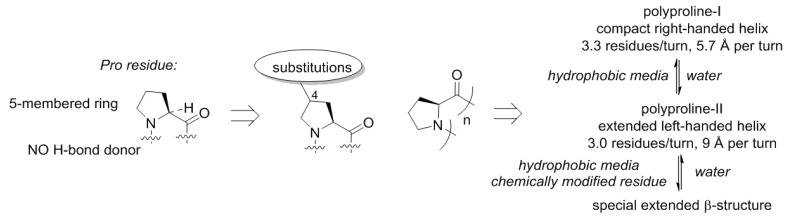
The structure diversity of the Proline World.

**Figure 7 ijms-20-05507-f007:**
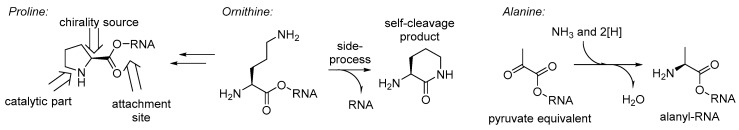
Chemical justification for the aminoacyl-RNA appearance in the GC-phase of the genetic code formation.

**Figure 8 ijms-20-05507-f008:**
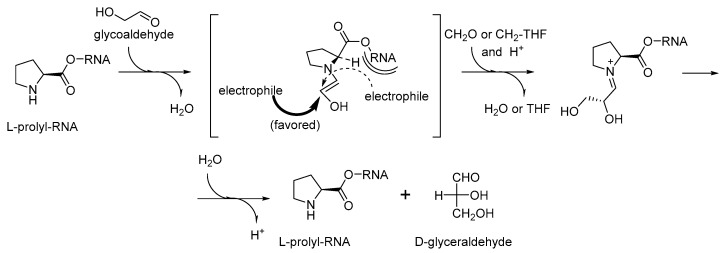
Possible condensation catalysis with chirality transfer on prolyl-RNA (as analogous to proline-catalyzed condensation [67,68]).

**Figure 9 ijms-20-05507-f009:**
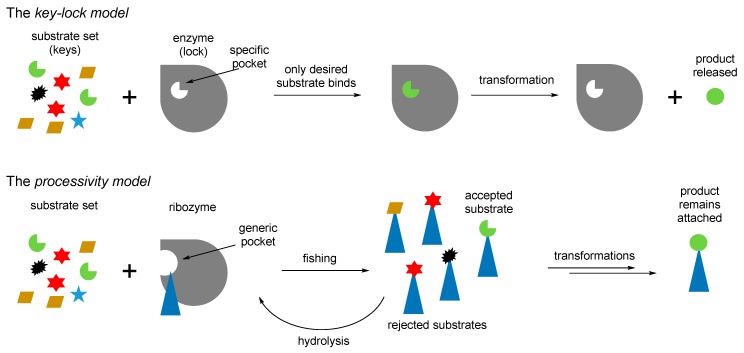
The key-lock model of enzyme chemistry versus the processivity model, which is likely to occur with the RNA mediated chemistry. The processivity model views the ribozyme as the Maxwell’s demon, a hypothetical creature that makes a system more complex through filtering. Many biological mechanisms operate this way [73].

**Figure 10 ijms-20-05507-f010:**
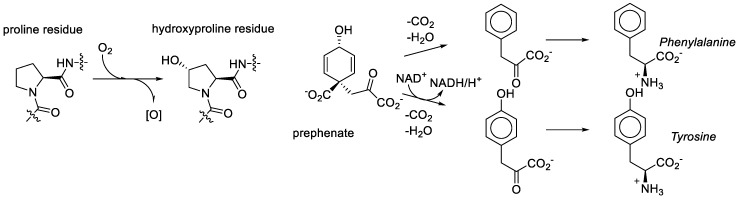
The oxidant requirements towards hydroxyproline and tyrosine are different.

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
