# Peer review of "The Alanine World Model for the Development of the Amino Acid Repertoire in Protein Biosynthesis"

_ijms, 2019, doi:10.3390/ijms20215507_

Round 1

Reviewer 1 Report

This manuscript describes an interesting scenario concerning how the existing 20 amino acids were chosen for protein component in the transition from the RNA world into a protein world during the origin and evolution of life. Their study led to what they claim the “Alanine World” model, a symbolic and a convincing phrase.

Based on the original theory by Hartman and Smith on the evolution of the genetic code, the GC-GCA-GCAU scheme, the authors found a correlation of this scheme with a hierarchy of protein folding. Accordingly, in the initial GC phase, where Ala, Gly, Pro, and perhaps Orn were the only ones coded, it was difficult to form a stable secondary structure. Once it entered GCA phase, those amino acids recruited such as Asp, Glu, Thr, Ser etc. were all substituted at beta-carbon of Ala and were able to form a stable alpha-helix based on Ala scaffold. It was this alpha-helix that enabled proteins to become stable in structure and became biologically functional. Finally in the GCAU phase, amino acids with hydrophobic side chains were recruited enabling proteins to penetrate into a cell membrane. This scenario is compatible with metabolism of amino acids and nucleotides biosynthesis. Thus, they came up with the conclusion that the stable structure of protein, and hence proteome, was possible on the core chemical scaffold of alanine. An alternative possible development of coded synthesis of proteins based on proline, the “Proline World”, is also discussed, which makes this manuscript even more interesting and insightful.

I certainly agree with every aspect of what authors insist in this manuscript. The authors claim that the choice of amino acids was not due to the structure of amino acids themselves but rather from the context of protein structure (amino acids do not equal proteins), which completely makes sense. These hypotheses should be widely taken into account when considering how and why particular coded protein synthesis started in the early form of life. Particularly, a discovery of a correlation between protein fold and the Hartman and Smith hypothesis is noteworthy. These perspectives will bring a new view on why the canonical 20 amino acids were brought into a coded protein synthesis. The manuscript will surely merit publication and would attract a lot of attentions from those interested in the topic and even from more general audience.

I may suggest that if an example of proline catalyzed condensation of sugars is provided as a figure between Figure 5 and 6 (Page 9, line 365; there are two Figure 5, one should be Fig. 6), perhaps it will make the manuscript more convincing.

Author Response

Reviewer 1:

I may suggest that if an example of proline catalyzed condensation of sugars is provided as a figure between Figure 5 and 6 (Page 9, line 365; there are two Figure 5, one should be Fig. 6), perhaps it will make the manuscript more convincing.

Reply:

We thank the referee for this valuable suggestion. We added Figure 8 to illustrate the chirality transfer argument. We also corrected the figure numbering.

Reviewer 2 Report

Alanine World

The authors should consider a number of relevant papers (Kim et al. 2018; Kim et al. 2019; Pak et al. 2018a; Pak et al. 2018b; Pak et al. 2017).

The central hypothesis of the current submission is flawed.

It makes little sense to evolve the code GC to GCA to GCAU.

The Alanine/Proline model requires building the code sector by sector. It is much more likely that the code evolved in large blocks (i.e. see references below). In this alternate scheme, previously added amino acids give up sectors to incoming amino acids but retain favored anticodons via selection.

The authors should consider the tRNA anticodon structure and the way the anticodon is read on the ribosome. The second position of the anticodon is the primary position. The third anticodon position is next most important followed by the wobble position, which is read ambiguously on the ribosome. Evolving the genetic code around the tRNA anticodon makes much better sense. Evolution of the anticodon-codon EF-Tu “latch” appears to parallel evolution of the genetic code.

tRNA drove mRNA evolution. In evolution of the genetic code, tRNA must have the central role. tRNA also drove evolution of the ribosome. Metabolism coevolved with coding.

There is a clear model for the sequence of primordial tRNA. The authors should consider this model.

Authors should consider tRNA structure and evolution.

Authors should consider Darwinian selection during steps of genetic code evolution.

Building a genetic code from a few occupied sectors poses problems for mRNA evolution. Essentially, mRNA sequence recognition must somehow converge with anticodon evolution.

These difficulties are overcome by filling in the code with a single amino acid (i.e. Gly) and allowing other amino acids to invade the code.

To this reviewer, the hypothesis underlying this paper is deeply flawed and should be re-thought. It is very difficult to imagine a Darwinian selection for establishment of tRNA, tRNAomes, the ribosome, mRNA and the genetic code working via the alanine world hypothesis.

References:

Kim Y, Kowiatek B, Opron K, Burton ZF (2018) Type-II tRNAs and Evolution of Translation Systems and the Genetic Code Int J Mol Sci 19 doi:10.3390/ijms19103275

Kim Y, Opron K, Burton ZF (2019) A tRNA- and Anticodon-Centric View of the Evolution of Aminoacyl-tRNA Synthetases, tRNAomes, and the Genetic Code Life (Basel) 9 doi:10.3390/life9020037

Pak D, Du N, Kim Y, Sun Y, Burton ZF (2018a) Rooted tRNAomes and evolution of the genetic code Transcription 9:137-151 doi:10.1080/21541264.2018.1429837

Pak D, Kim Y, Burton ZF (2018b) Aminoacyl-tRNA synthetase evolution and sectoring of the genetic code Transcription 9:205-224 doi:10.1080/21541264.2018.1467718

Pak D, Root-Bernstein R, Burton ZF (2017) tRNA structure and evolution and standardization to the three nucleotide genetic code Transcription 8:205-219 doi:10.1080/21541264.2017.1318811

Author Response

Reviewer 2:

Alanine World

The authors should consider a number of relevant papers (Kim et al. 2018; Kim et al. 2019; Pak et al. 2018a; Pak et al. 2018b; Pak et al. 2017).

The central hypothesis of the current submission is flawed.

It makes little sense to evolve the code GC to GCA to GCAU.

The Alanine/Proline model requires building the code sector by sector. It is much more likely that the code evolved in large blocks (i.e. see references below). In this alternate scheme, previously added amino acids give up sectors to incoming amino acids but retain favored anticodons via selection.

The authors should consider the tRNA anticodon structure and the way the anticodon is read on the ribosome. The second position of the anticodon is the primary position. The third anticodon position is next most important followed by the wobble position, which is read ambiguously on the ribosome. Evolving the genetic code around the tRNA anticodon makes much better sense. Evolution of the anticodon-codon EF-Tu “latch” appears to parallel evolution of the genetic code.

tRNA drove mRNA evolution. In evolution of the genetic code, tRNA must have the central role. tRNA also drove evolution of the ribosome. Metabolism coevolved with coding.

There is a clear model for the sequence of primordial tRNA. The authors should consider this model.

Authors should consider tRNA structure and evolution.

Authors should consider Darwinian selection during steps of genetic code evolution.

Building a genetic code from a few occupied sectors poses problems for mRNA evolution. Essentially, mRNA sequence recognition must somehow converge with anticodon evolution.

These difficulties are overcome by filling in the code with a single amino acid (i.e. Gly) and allowing other amino acids to invade the code.

To this reviewer, the hypothesis underlying this paper is deeply flawed and should be re-thought. It is very difficult to imagine a Darwinian selection for establishment of tRNA, tRNAomes, the ribosome, mRNA and the genetic code working via the alanine world hypothesis.

Reply:

There are numerous theories for the emergence and development of the protein biosynthesis. As mentioned in the Editorial comment, some contradict our scheme, others (like co-evolution, co-enzyme handle, neutral theory) clearly agree with our model. The order of the codons is not our invention, however. This is a scheme developed previously by Hartman and Smith. The criticism of the referee should be addressed to their articles. In our manuscript, we do not develop, but interpret it in terms of peptide/protein chemistry. The authors are not RNA chemists. We are peptide/protein chemists. Therefore, we cannot ‘consider tRNA structure and evolution’ as suggested by the referee. We can interpret the existing scheme. The schemes suggested in the proposed papers make little sense in terms of protein chemistry. E.g. a suggestion that original polypeptide synthesis targeted synthesis of polyglycine for the cell walls is supported by an example of one organism, while the cell wall synthesis is clearly known to be non-ribosomal. 

Moreover, it is also not clear whether the evolution of the genetic code was actually a Darwinian process. It is highly questionable whether the evolution phases before LUCA can be perceived as Darwinian. In our analysis we avoid discussion of this issue, and treat it mainly as a molecular evolution process, where each next step starts accidentally and leads to increase of the molecular complexity, and local molecular versatility. In the revised manuscript we added a few sentences to stress on this. 

We hope that the referee understands that we cannot completely re-write manuscript following a completely different scheme. In the future we would be glad to consider variety of schemes, and compare their vitality in terms of the protein chemistry. However, this is not the point of the presented work. 

Reviewer 3 Report

This paper presents a new hypothesis on a possible stage in the development of life, the transition from a possible RNA World to a protein world.  I believe that such hypothesis papers must be published because they open the field of discussion even if a referee (like me) may or may not agree with the proposed hypothesis.

It should remain clear however that this is a hypothesis, and must open the debate. For example, the last sentence (we are convinced...) of the text is far too affirmative and should be replaced or followed by an open conclusion, calling for debate.

The fact that the "RNA world" is no more than a hypothesis should be clearly stated in the introduction.

I do not think the term "alanine world" reflects the truth of what is being presented. The very fact of considering that a first group of amino acids is "derived from alanine" is debatable. This is not based on today's metabolic truth, and that questions the given title.

It is very odd to prefer ornithine to lysine.

The part concerning the "Proline world" is problematic. If I understand correctly, it describes a form of life that does not exist but which, according to the authors, could have existed. What is the point? Is not that excessive? The authors should at least seriously tighten parts 6 and 7.

Some sentences say really trivial things and could be omitted: line 135 to 138 ; line 453 "the process is also known as protein translation or protein coding". Others are surprisingly written: line 179, "the repertoire was evolutionary 'convicted' to undergo an expansion" ???

Overall, however, I think this article can be published after being edited. Perhaps the MDPI journal "Life" would be more appropriate for its publication.

Author Response

Reviewer 3:

It should remain clear however that this is a hypothesis, and must open the debate. For example, the last sentence (we are convinced...) of the text is far too affirmative and should be replaced or followed by an open conclusion, calling for debate.

Reply:

The sentence were removed, and the argument was reformulated.

Reviewer 3:

The fact that the "RNA world" is no more than a hypothesis should be clearly stated in the introduction.

Reply:

We call it a hypothesis throughout the article. We never said that it is ‘more than a hypothesis’. 

Reviewer 3:

I do not think the term "alanine world" reflects the truth of what is being presented. The very fact of considering that a first group of amino acids is "derived from alanine" is debatable. This is not based on today's metabolic truth, and that questions the given title.

Reply:

By ‘alanine derivatives’ we call structures that share same core architecture, we do not mean metabolic derivatives. In the revised manuscript we added a sentence on p.5 stressing on this.

Reviewer 3:

It is very odd to prefer ornithine to lysine.

Reply:

This is odd in the Protein World, therefore the Protein World rejected ornithine structure eventually. But in the RNA World the protein biosynthesis was not an intended purpose of the amino acid recruitment, therefore the logic was different. There is a whole part of discussion that explains the choice of ornithine. Ornithine produces proline, which is a good condensation catalysts. Lysine would produce pipecolic acid, which is a much worse condensation catalysts. 

Reviewer 3:

The part concerning the "Proline world" is problematic. If I understand correctly, it describes a form of life that does not exist but which, according to the authors, could have existed. What is the point? Is not that excessive? The authors should at least seriously tighten parts 6 and 7.

Reply:

We cannot claim the Alanine World without explaining what it is. The best it can be done by showing an alternative. An alternative therefore deserves as much attention as the Alanine World itself. If the reader is not interested in this description, they can simply skip the section. However, we’re convinced that scientific concepts should be conceived through the analysis of the alternative schemes. This is why we gave so much attention to the Proline World.

Reviewer 3:

Some sentences say really trivial things and could be omitted: line 135 to 138 ; line 453 "the process is also known as protein translation or protein coding". Others are surprisingly written: line 179, "the repertoire was evolutionary 'convicted' to undergo an expansion" ???

Reply:

The sentences were removed.

Reviewer 3:

Overall, however, I think this article can be published after being edited. Perhaps the MDPI journal "Life" would be more appropriate for its publication.

Reply:

This article was invited to the IJMS special issue "Origins of Translation', and this is exactly the topic of the article. 

Round 2

Reviewer 2 Report

Alanine World Hypothesis

Authors should model genetic code evolution starting with the core feature and work out. The core feature is tRNA (Kim et al. 2019).

Why do the authors think that some current metabolic pathways mimic primordial pathways? Is this a reasonable assumption?

Line 102: the authors discuss “chemical logic”. The chemical logic behind evolution of the genetic code is tRNA structure, the tRNA anticodon and functions in translation (Kim et al. 2019).

Concentrating on amino acid metabolism is misleading. Evolution of amino acids is largely a parallel process to the evolution of the genetic code.

Aminoacyl-tRNA synthetase evolution tracks the population of the genetic code sectors (Kim et al. 2019). ValRS-IA is an early entry into the code. 

Here is a new paper that seems relevant and perhaps should be cited:

https://www.sciencemagazinedigital.org/sciencemagazine/04_october_2019_Main/MobilePagedReplica.action?u1=02524538&utm_source=newsletter&utm_medium=email&utm_campaign=TXSCI2191003004&pm=1&folio=76#pg82 (Hud and Fialho 2019)

Authors use “since” to mean “because”. This is poor English usage.

Authors use contractions: i.e. “don’t” should be “do not” or something else.

There are many errors in English presentation that should be corrected.

Gly and Asp are necessary to generate nucleic acids. Why is ornithine brought into the code before Asp?

The code evolved around tRNA before mRNA, so the code should be considered as an anticodon code as well as a codon code (Kim et al. 2019).

It is clear that U was available when tRNA became available. Otherwise the U-turn after a U residue to form the anticodon loop and the T loop would not be possible (Pak et al. 2017).

The authors have not considered tRNA in any meaningful way in their model.

The authors apply a mRNA/codon-centric view of genetic code evolution. This is a mistake.

The model as proposed is inconsistent with aaRS evolution. Class I aaRS are derived from ValRS-IA, which is derived from GlyRS-IIA by refolding (Kim et al. 2019). From ValRS-IA comes IleRS-IA, LeuRS-IA and MetRS-IA. This is inconsistent with the GCàGCAàGCAU model.

Protein folding is mostly a separate problem (a parallel problem) compared to code evolution.

Modeling evolution of the genetic code should start with tRNA and work out (Kim et al. 2019).

References:

Hud NV, Fialho DM (2019) RNA nucleosides built in one prebiotic pot Science 366:32-33 doi:10.1126/science.aaz1130

Kim Y, Opron K, Burton ZF (2019) A tRNA- and Anticodon-Centric View of the Evolution of Aminoacyl-tRNA Synthetases, tRNAomes, and the Genetic Code Life (Basel) 9 doi:10.3390/life9020037

Pak D, Root-Bernstein R, Burton ZF (2017) tRNA structure and evolution and standardization to the three nucleotide genetic code Transcription 8:205-219 doi:10.1080/21541264.2017.1318811

Author Response

Reviewer 2:

Alanine World Hypothesis

Authors should model genetic code evolution starting with the core feature and work out. The core feature is tRNA (Kim et al. 2019).

Line 102: the authors discuss “chemical logic”. The chemical logic behind evolution of the genetic code is tRNA structure, the tRNA anticodon and functions in translation (Kim et al. 2019).

Reply:

We do not agree with the referee. The purpose of translation is polymerization of amino acids into proteins. Therefore, amino acids and proteins are at the center of the translation process. Protein is the outcome of it. The m/r/tRNA is a mean of translation, but not its purpose. Our model puts proteins in the center of attention, and suggests how the protein folding features evolved in the course of the amino acid repertoire expansion.

Reviewer 2:

Why do the authors think that some current metabolic pathways mimic primordial pathways? Is this a reasonable assumption?

Reply:

We assume that metabolic pathways were inherited because we think that the metabolism was a continuous process. I.e. if a certain metabolite was produces in the RNA World phase, proteins took over its production by mimicking catalysis of the individual metabolic steps. We believe, this is a reasonable assumption.

Reviewer 2:

Concentrating on amino acid metabolism is misleading. Evolution of amino acids is largely a parallel process to the evolution of the genetic code.

Reply:

The availability of the amino acids and their metabolic production was essential for their recruitment into the genetic code. If there is no amino acid available, it cannot be taken to translation. If there is no metabolic production of an amino acids, it cannot remain in the translation process. This is simple logic.

Reviewer 2:

Aminoacyl-tRNA synthetase evolution tracks the population of the genetic code sectors (Kim et al. 2019). ValRS-IA is an early entry into the code.

Reply:

The evolution of aaRA have a limited relevance to the understanding of the genetic code. It is because the information about aaRS and their evolution is taken from genes. The genetic code assigns genes with meanings. Before the genetic code, the genes were not having their full meanings; therefore, the analysis is of genetic code evolution via the analysis of aaRS is not adequate, although it can provide some hints. There is a whole part of the discussion in section 2, which explains this argument.

The ValRS could be an early or late invention, it does not matter. The addition of Val to the genetic code is completely unrelated to this. It is because the charging of the amino acids was invented before there were proteins, in the RNA World phase. aaRS is a protein. They took over this function, but the function existed before. The RNA World started producing proteins, when the proteins didn’t exist. This is what the RNA World hypothesis says.

Reviewer 2:

Here is a new paper that seems relevant and perhaps should be cited:

https://www.sciencemagazinedigital.org/sciencemagazine/04_october_2019_Main/MobilePagedReplica.action?u1=02524538&utm_source=newsletter&utm_medium=email&utm_campaign=TXSCI2191003004&pm=1&folio=76#pg82 (Hud and Fialho 2019)

Reply:

The paper is from prebiotic chemistry, which is a bottom-up reconstruction of life evolution. Our approach is a top-down deconstruction. We clearly state this in the concluding section.

Reviewer 2:

Authors use “since” to mean “because”. This is poor English usage.

Authors use contractions: i.e. “don’t” should be “do not” or something else.

There are many errors in English presentation that should be corrected.

Reply:

We would like to keep the contraction for the style purposes. “Since” was used three times, where it can be substituted with “because”, that does not make much of a difference for the narrative.

There are indeed typos in the revised version. We tried our best to remove them by a few rounds of careful reading. We apologize for any inconveniences. 

Reviewer 2:

Gly and Asp are necessary to generate nucleic acids. Why is ornithine brought into the code before Asp?

Reply:

Asp is negatively charged, so are polyanionic RNA bodies. A peptide containing Asp would swim away, and will be lost. The initial set of amino acids must contain positively charged residues in order to adhere to the RNA bodies, and in this way create initial feedback for the evolution. We incorporated this argument into section 3.

Reviewer 2:

The code evolved around tRNA before mRNA, so the code should be considered as an anticodon code as well as a codon code (Kim et al. 2019).

It is clear that U was available when tRNA became available. Otherwise the U-turn after a U residue to form the anticodon loop and the T loop would not be possible (Pak et al. 2017).

The authors have not considered tRNA in any meaningful way in their model.

The authors apply a mRNA/codon-centric view of genetic code evolution. This is a mistake.

Reply:

Another point of view is not a mistake. It is just different. The evolution of life is a complex process, which involved the contribution of a number of physical and chemical principles: availability of organic material, metabolic feasibility, RNA structures, protein fold, etc. Each provides its own unique perspective on the problem of the primordial life development. We are peptide/protein chemists; therefore, we wrote the article from the perspective of protein chemistry.

Reviewer 2:

The model as proposed is inconsistent with aaRS evolution. Class I aaRS are derived from ValRS-IA, which is derived from GlyRS-IIA by refolding (Kim et al. 2019). From ValRS-IA comes IleRS-IA, LeuRS-IA and MetRS-IA. This is inconsistent with the GCàGCAàGCAU model.

Reply:

As mentioned above, because aaRSs are proteins, their sequences and structures cannot answer the question how the protein synthesis emerged.

Reviewer 2:

Protein folding is mostly a separate problem (a parallel problem) compared to code evolution.

Reply:

Protein is the result of the evolution of the genetic code. This is why, the evolution of the proteins, their folding and functions, can be viewed as providing driving process behind the genetic code development.  Analysis of the protein fold development is reasonable.

Reviewer 2:

Modeling evolution of the genetic code should start with tRNA and work out (Kim et al. 2019).

Reply:

This article as well as few other suggested by the reviewer 2 earlier state that the evolution was going around proto-tRNA. The papers never justify this statement. For example, one of the papers, suggested by the reviewer (Kim et al., doi: 10.1080/21541264.2017.1318811) says that 75 nt core of tRNA is unaltered since LUCA. However, the protein translation developed before LUCA. There is no other justification provided.  

That tRNA evolution should be considered as a driving process is purely an opinion of the authors of the suggested articles, and we have a right not to adhere to it. 

Reviewer 3 Report

Taking into account the answers given to my questions/commentaries I now recommend acceptance of this manuscript.

Author Response

Reviewer 3:

Taking into account the answers given to my questions/commentaries I now recommend acceptance of this manuscript.

Reply:

We thank our reviewer for their time invested in our manuscript and the suggestions.